# Future Directions in the Treatment of Osteosarcoma

**DOI:** 10.3390/cells10010172

**Published:** 2021-01-15

**Authors:** Alannah Smrke, Peter M. Anderson, Ashish Gulia, Spyridon Gennatas, Paul H. Huang, Robin L. Jones

**Affiliations:** 1Sarcoma Unit, Royal Marsden Hospital, 203 Fulham Road, London SW3 6JJ, UK; alannah.smrke@rmh.nhs.uk (A.S.); spyridon.gennatas@rmh.nhs.uk (S.G.); 2Pediatric Hematology Oncology and Bone Marrow Transplantation, Cleveland Clinic R3 Main Campus, 9500 Euclid Avenue, Cleveland, OH 44195, USA; andersp@ccf.org; 3Orthopedic Oncology Services, Department of Surgical Oncology, Tata Memorial Hospital, HBNI, Mumbai 400012, India; aashishgulia@gmail.com; 4The Institute of Cancer Research, 237 Fulham Road, London SW3 6JB, UK; Paul.Huang@icr.ac.uk

**Keywords:** osteosarcoma, chemotherapy, MAP, immunotherapy, genomic heterogeneity, adolescent and young adult, patient outcomes

## Abstract

Osteosarcoma is the most common primary bone sarcoma and is often diagnosed in the 2nd–3rd decades of life. Response to the aggressive and highly toxic neoadjuvant methotrexate-doxorubicin-cisplatin (MAP) chemotherapy schedule is strongly predictive of outcome. Outcomes for patients with osteosarcoma have not significantly changed for over thirty years. There is a need for more effective treatment for patients with high risk features but also reduced treatment-related toxicity for all patients. Predictive biomarkers are needed to help inform clinicians to de-escalate or add therapy, including immune therapies, and to contribute to future clinical trial designs. Here, we review a variety of approaches to improve outcomes and quality of life for patients with osteosarcoma with a focus on incorporating toxicity reduction, immune therapy and molecular analysis to provide the most effective and least toxic osteosarcoma therapy.

## 1. Introduction

Osteosarcoma is the most common primary sarcoma of bone [1] whose cell of origin produces bone or osteoid and varying amounts of cartilaginous matrix and fibrous tissue. It most commonly arises within the metaphysis of growing long tubular bones [2]. Uncommonly, 6%–10% of osteosarcomas present in the craniofacial bones [3]. Annual incidence is estimated at 2–4 patients per million [2,4]. Males are more likely to be affected than females [1,4,5]. It can occur at any age, however, peak incidence is in the second and third decade of life [2]. Nearly one quarter of patients present with metastatic disease, with the lungs being the most common site [5]. In older patients, axial skeleton primary, larger tumour size and lower socio-economic status are associated with a higher likelihood of metastatic disease [5]. Pathologically, high-grade osteosarcomas are divided into osteoblastic, chondroblastic, fibroblastic, teleangiectatic, giant-cell rich, small cell and sclerosing [6]. On the basis of a large cohort of 570 patients, osteoblastic is shown to be the most common subtype (71%) followed by chondroblastic (10%) and fibroblastic (9%) [6].

Prior to the 1970s, patients with osteosarcoma were treated with surgery alone, with event free survival (EFS) estimated at 20% [7]. A randomised trial of 36 patients treated with multi-agent adjuvant chemotherapy revolutionised the care of patients with localised disease [7]. Patients under 30 years of age were treated with a complex schedule of cyclophosphamide, bleomycin, dactinomycin, high dose methotrexate (HDMTX), doxorubicin and cisplatin within two weeks of primary resection or active surveillance alone. Two-year relapse free survival was significantly improved; 17% in the active surveillance arm compared to 66% with adjuvant chemotherapy [7]. In subsequent studies, use of doxorubicin-cisplatin alone has been shown to have equivalent survival compared to multi-agent chemotherapy [8]. There is limited randomised data for the benefit of HDMTX. HDMTX has the advantage of less myelosuppression than doxorubicin-cisplatin but can be difficult to give to adult patients because of delayed clearance and mucositis. Recently, presented work by the Italian Sarcoma Group has suggested a significant EFS and overall survival (OS) benefit for the treatment of localised extremity osteosarcoma with 10 cycles (cumulative dose 120 g/m^2^) compared to 5 cycles (cumulative dose 60 g/m^2^) [9] of methotrexate with a doxorubicin-cisplatin backbone. Nonetheless, the international standard perioperative regimen for patients under 40 years of age is accepted to be high-dose methotrexate, doxorubicin and cisplatin (MAP) [10,11].

The contemporary EFS and OS for patients with osteosarcoma are unfortunately largely unchanged [4,12,13], with a stagnation in the discovery of novel, effective therapies. Here, we review ongoing and future strategies to improve outcomes of patients with osteosarcoma.

## 2. The Role of Tumour Size and Pathological Response

Both tumour size and pathological response have been shown to influence metastasis-free survival in a large cohort of patients with localised osteosarcoma [14]. A nomogram predicting metastasis-free survival based on tumour size (AJCC stage) and percent necrosis has been published [14]. Tumours >371 cm^3^ (approximate volume of a ellipsoid is 4/3 r^3^) have two times greater incidence of lung metastasis [15]. In addition, tumour volume increase during neoadjuvant chemotherapy is also predictive of local recurrence [16].

The extent of the pathologic response to neoadjuvant chemotherapy using tumour cell necrosis emerged as a predictor of survival from early cohorts of patients with osteosarcoma treated with chemotherapy [8,17,18]. Patients with 90% or greater necrosis are defined as good risk and have a five-year disease-free survival (DFS) of 75%–93%. Patients with less than 90% necrosis are considered poor risk with a five-year DFS of 45%–67% [8,9,10,17,18]. In reality, percent necrosis is probably a continuous variable [19], however, the dichotomous cut point of 90% remains firmly established in the literature.

Beyond tumour size and pathological response, immune factors may improve survival. Similar to percent necrosis, absolute lymphocyte count (ALC) recovery after the initial cycle of chemotherapy has been shown to correlate to improved OS in paediatric patients with osteosarcoma [20,21,22]. In addition, a retrospective series of patients who underwent surgery for their osteosarcoma demonstrated that patients who developed a deep infection within 12 months of surgery had a significantly improved survival [23]. It is quite possible that the immune activation associated with infection may contribute to improved outcomes in this subset of patients. Finally, there is evidence that mifamurtide, a macrophage activator approved for osteosarcoma in Europe, Israel, Korea, Mexico and Taiwan, and via compassionate access (currently via single patient IND regulatory mechanism using FDA information available at ONCProjectfacilitate@fda.hhs.gov) in the USA, can improve outcomes for patients [24,25,26,27,28]. Taken together, this provides initial rationale that immunotherapy may have a role in the treatment of osteosarcoma. One of us (PMA) uses this information correlating immune variables with durable osteosarcoma responses to help patients and caregivers re-focus the more than two-thirds of systemic therapy that is given after limb salvage surgery. Such changes include reducing treatment related toxicity, such as nausea, and mucositis [29] and improving immune function via better nutrition [29]. There is also an increased focus on outpatient therapy using chemotherapy pumps with supportive medication regimens during this period [30]. For patients with metastatic disease, local control of metastases (surgery, SBRT and/or cryoablation) can be considered to reduce disease burden. Finally, focusing on holistic care for patients and caregivers using strategies such as a virtual visit tool and focusing on ‘making things better’ is key to improve the quality of life in treatment [31].

Beyond pathological response, recent work from Palmerini et al. was designed to modify therapy based on the expression of P-glycoprotein (Pgp) [9]. Overexpression of Pgp (predominantly measured by immunohistochemistry) has been shown in a meta-analysis to correlate with more aggressive disease [32] but not histological response to neoadjuvant chemotherapy [32,33]. To explore treatment modifications based on Pgp expression, the trial modified treatment for patients with high-grade extremity osteosarcoma is based on both histological response to neoadjuvant MAP and Pgp expression. Patients without Pgp overexpression received adjuvant MAP irrespective of pathological response. Patients with Pgp overexpression were treated with miframurtide, and either with MAP if there was a good response or ifosfamide if there was a poor response. There was no difference in EFS based on Pgp expression in this cohort. It remains to be seen whether the differing treatment arms based on Pgp expression removed the impact of Pgp expression on prognosis, or Pgp expression alone may not be prognostic in osteosarcoma.

Further translational work to subdivide osteosarcoma into prognostic and predictive risk groups will be key to understanding its heterogeneous behaviour and improving treatment outcomes. This has been explored in a microarray based analysis of diagnostic biopsy samples from 30 paediatric patients with osteosarcoma [34]. When samples were grouped compared to subsequent response to neoadjuvant MAP treatment, differing expression profiles were seen. There were 910 genes differentially expressed on the basis of the extent of necrosis, and with further refinement, there were 104 genes which met significance [34]. Osteosarcomas with poor response to MAP were associated with increased expression of genes involved in osteoclast promotion, bone resorption, cell migration and apoptosis resistance, while genes which regulate cell motility were reduced [34]. It is also important to perform such similar analyses post MAP chemotherapy to understand if this expression profile changes with chemotherapy exposure. Further, larger scale work is required to provide insights into the differing biology of osteosarcomas at diagnosis and pathological response to neoadjuvant chemotherapy. This will enable the identification of prognostic groups at diagnosis and potential rationale targets for novel modifications to treatment. However, in routine clinical practice, we remain limited to pathological response as the most well studied prognostic biomarker.

## 3. Tailoring Treatment for Patients with Good Pathological Response

For patients within the good pathological response group (90% or greater necrosis), there may be opportunities to further optimise therapy. One option is improving outcomes through the de-escalation of therapy and reduction in treatment-related toxicity. Given that percent necrosis is likely a continuous variable [19], one could consider a more conservative cut off of, such as >95% necrosis for initial de-escalation trials. The serious acute and long term effects patients experience from MAP treatment are well described and include cardiotoxicity, neurotoxicity, ototoxicity, nephrotoxicity, infertility and secondary neoplasms [35]. It may be that fewer cycles or chemotherapy drugs are required to obtain the same pathological response within the good risk group. Exploring such a strategy could involve a randomised trial with two cycles of neoadjuvant MAP, and multiple adjuvant arms, such as mifamurtide or active surveillance, MAP, HDMTX monthly and doxorubicin-cisplatin alone.

## 4. Tailoring Treatment for Patients with Poor Pathological Response

On the basis of the contemporary EURAMOS-1 trial, over half of all patients treated with neoadjuvant chemotherapy have a poor response to MAP chemotherapy [10]. The paradigm of modifying adjuvant therapy based on pathologic response to neoadjuvant therapy has been successful in the treatment of other cancers, namely breast cancer [36,37]. While we acknowledge the differing biology of osteosarcoma and breast cancer, conceptually, it is important that trials explore such modifications of adjuvant treatment. In an attempt to improve outcomes within the poor risk group, modifying chemotherapy based on pathological response was performed in EURAMOS-1 trial [10]. In this international phase III randomised trial, 2260 patients with localised or metastatic osteosarcoma (all metastatic sites needed to be operable at registration) were registered and received two cycles of doxorubicin-cisplatin and no more than six courses of methotrexate pre-operatively. Patients were then randomised 1:1 to adjuvant MAP or MAP plus ifosfamide and etoposide (MAPIE). Unfortunately, there was no significant difference in EFS between groups, and MAPIE was more toxic [10]. Although MAPIE was disappointing in EURAMOS, it may be related to low efficacy of etoposide [38,39] and low intensity/dose-density of ifosfamide + mesna given in the EURAMOS trial. Ifosfamide + mesna can indeed be given with excellent effectiveness against osteosarcoma with high dose-density (two weeks each month) and intensity (14 g/m^2^ per cycle) and less toxicity (e.g., less CNS, renal and thrombocytopenia) when given as a continuous infusion [40,41,42,43]. Furthermore, ifosfamide + mesna as a continuous infusion can be given in the outpatient clinic, a welcome respite from frequent hospitalization associated with MAP. The addition of such a schedule of ifosfamide to upfront treatment of osteosarcoma warrants further exploration. However, we acknowledge that currently, pathologic response information is frustrating for patients and clinicians, as we are in a quagmire of having worrying information without any evidence-based options to improve outcomes for patients.

Beyond ifosfamide and etoposide, the rational selection of agents to further modify adjuvant chemotherapy remains difficult. There are limited active treatments for patients with metastatic disease, including ifosfamide-etoposide (IE) [44], single agent ifosfamide [40,42,43], regorafenib [45], gemcitabine-docetaxel [46], cabozantinib [47], pazopanib [48] ± SBRT [49] and radium-223 alone or in combination [50,51]. Given the modest response rates to therapies beyond ifosfamide ± etoposide, a thoughtful, likely translational, approach is needed to identify new therapies with a high likelihood of success. This hinges on large scale international partnerships to understand the genomic alterations that occur within osteosarcoma and its subtypes. In addition, translational work is also needed to identify predictive biomarkers at diagnosis is needed to help preselect good and poor risk patients to improve clinical trial design. For example, if clinicians knew upfront that standard chemotherapy was ineffective as their patient had poor risk osteosarcoma, then such patients may be considered for upfront surgery and early clinical trial enrolment. This group of patients may be attractive for trials to improve efficacy of immune therapy against osteosarcoma. Options could include anti-CD47 + mifamurtide, anti-CD47 + anti-osteosarcoma directed antibodies [52] or cellular therapy (CAR-T-cells).

## 5. Maintenance Therapy

Another option to improve outcomes for patients with osteosarcoma is maintenance therapy. This approach has been successful in patients with rhabdomyosarcoma [53]. To test this hypothesis, patients with good response to neoadjuvant chemotherapy in the EURAMOS-1 trial randomised patients with good pathological response to maintenance pegylated interferon alfra-2b versus active surveillance [54]. Unfortunately, there was no difference in the three-year EFS with the addition of interferon. While these results were disappointing, the maintenance approach remains an important consideration for future trial design.

In terms of other approaches, outpatient metronomic approaches such as those seen with rhabdomyosarcoma [53], particularly for patients with poor responding tumours, could be considered in future trial designs. The difficulty lies in the selection of the ‘active’ agent. There is evidence that bisphosphonates, which inhibit osteoclasts and therefore bone resorption, can have inhibitory effects in vitro [55]. Pamidronate given monthly for 12 months has been added to MAP chemotherapy in a cohort of 40 patients. Pamidronate was safe, with EFS and OS in the cohort similar to published ranges for patients treated with chemotherapy alone [56]. There were no cases of osteonecrosis of the jaw, and other common side effects (ototoxicity and nephrotoxicity) were observed at similar rates to large pediatric cooperative group trials without pamidronate [56]. Given the differential expression of osteoclast proliferative genes that has been shown in poor risk osteosarcoma [34], a randomised maintenance design with a bisphosphonate may be warranted in future clinical trials.

## 6. The Role of DNA and RNA Analysis

It is known that clinical behaviour and outcome of osteosarcoma varies on the basis of the primary site, age and whether it is primary versus radiation-associated [3,57]. This variability suggests that understanding the biologic heterogeneity amongst osteosarcomas may be key to improve patient outcomes. Osteosarcomas have been shown to have significant variation in somatic copy number alteration (SCNA) which is thought to be the driver of oncogenesis and metastasis [58]. Whole genome sequencing (WGS) of 30 tumour samples from patients at diagnosis, post chemotherapy and at recurrence has revealed the amplification of *MYC* (39%), *CCNE1* (33%), *VEGFA* (23%), *CDK4* (11%), gain of *AURKB* (13%) and *PTEN* loss (56%) [58]. Though the sample size is small, these results highlight the heterogeneity of genetic alterations in osteosarcoma. The same group elegantly treated patient-derived xenografts (PDX) with drugs which both targeted (i.e., palbociclib for CDK4 amplification) and did not target the SCNA seen in the xenograft [58]. There was a response only in PDXs treated with drugs which targeted their SCNA [58].

A next generation sequencing (NGS) panel (MSK-IMPACT) was performed on samples from 72 patients with osteosarcoma [59]. Many different SCNA amplifications were seen, including in *VEGFA* (27%), *MDM2* (15%), *CDK4* (13%), *KIT* (15%), *KDR* (15%), *PDGFRA* (18%) and *MYC* (8%). Of the patients tested, 21% of patients had a finding which translated to the potential use of a currently available, but off-label drug or one in clinical trial testing [59]. Thus, NGS was able to identify a potential therapeutic target for a clinically meaningful number of patients. These results show promise that neoadjuvant or adjuvant approaches which tailor a component of therapy to specific genome alterations may be possible as WGS and NGS becomes more widely available in the oncology community.

For patients with metastatic disease, particularly those with chemo-resistant/refractory disease, DNA and RNA analysis is often considered to provide information regarding further potential therapeutic targets. Comprehensive molecular profiling of two such patients was undertaken by Subbiah et al. [60]. While unique alterations leading to a targeted treatment were identified in both patients, neither patient had a clinical benefit from molecularly-selected therapy. This suggests that precision medicine is possible, however, this study also highlights that work is still required to analyse the complex alterations within osteosarcoma to identify the putative drivers of oncogenesis.

Given the inherent heterogeneity of osteosarcoma, the further integration of precision medicine-based approaches into standard of care treatment is required. This will most importantly provide evidence for access to non-standard treatments which may lead to clinically meaningful benefit for patients. Such efforts must be collaborative to further global understanding osteosarcoma biology and identify novel therapeutic targets, such as ‘Target Osteosarcoma’ [61]. In addition, some commercial platforms (e.g., Tempus) also provide not only tumour mutation analysis and RNA expression data but also germline data. This is particularly important in osteosarcoma because some patients have germ-line p53 mutations [62,63]. It is paramount that patients who have WGS/NGS-directed therapy are either enrolled in well-designed clinical trials or participate in a registry such as “Count Me In” [64]. Furthermore, such trials and registries represent an important partnership between pharmaceutical companies, clinicians, and patients and caregivers. These partnerships have the potential to expand datasets and enable patients to access potentially beneficial therapies which are unavailable, but need data to become available on a case-by case basis for a rare indication, or are prohibitively expensive off of a clinical trial.

It is clear that future osteosarcoma therapy will require continued reliance on the ‘bench to bedside’ approach. For example, the proto-oncogene c-SRC (Src) is a non-receptor tyrosine kinase which is involved in cell growth and survival. Src has been shown to be increased in osteosarcoma tissues and cell lines [65,66]. Treatment of osteosarcoma cell lines and cell culture-derived xenografts with Src inhibitors has been shown to impair cell viability, induce apoptosis and decrease tumour mass [67]. Analysis also suggested that cytoplasmic Src localisation in human osteosarcoma samples may be associated with long term survival [68]. Together, this data formed strong pre-clinical evidence that Src inhibitors may have activity in osteosarcoma. A phase 2.5 double-blind, placebo-controlled trial of the oral Src inhibitor sarcatinib was conducted in patients with osteosarcoma with pulmonary metastasis that had or had not been resected [69]. Even with biological rationale, there was unfortunately no difference in OS in the sarcatinib and placebo arms [69]. Interestingly, pre-clinical work using dasatinib, another src inhibitor, also showed that while Src activity was inhibited by dasatinib in vitro, this did not prevent the development of pulmonary metastasis in mouse models [70]. While these results once again highlight the molecular complexity of osteosarcoma, and while the results are disappointing, this example is the type of translational collaboration that we should strive for to improve outcomes in patients with osteosarcoma.

## 7. Does Genomic Heterogeneity Suggest Efficacy of Immunotherapy?

Recent work has shown the expression of PD-L1 and tumour-infiltrating T cells in osteosarcoma patient specimens [71]. PDL-1 and PD-1 expression has been shown to be highest in pre-treatment biopsies but also importantly present even in decalcified specimens, making this a potentially useful biomarker for further study [72]. A meta-analysis has shown that PDL-1 and PD-1 expression may correlate with development of metastases and risk of mortality [73]. Congruently, PDL-1 expression is higher in patients with metastatic compared to localised disease [71]. On the basis of transcriptomic analysis, low expression of immune related genes has been shown to correlate with poor patient outcomes [74]. Immune infiltration score estimates from a group of human osteosarcoma samples have demonstrated that the majority of human osteosarcoma tumours have an average immune infiltration [75]. Average immune infiltration is thought to correlate with a lack of efficacy of immunotherapy. Interestingly, high immune infiltration scores (as seen in melanoma and lung cancer) are associated with clinical benefit from immunotherapy and were seen on a minority (8%–10%) of osteosarcoma samples [75]. In patients with localised disease, patients with CD8 and cytotoxic T cell expression within the tumour microenvironment correlated with improved survival [76]. Interestingly, chemotherapy did not change immune cell expression, once again suggesting there may be a role for immunotherapy to generate a more favourable immune rich microenvironment. Taken together, this suggests a potential role for immunotherapy for at least a subset of patients with osteosarcoma [77].

Within immunotherapy trials of multiple paediatric tumour subtypes, response was seen in one (n = 1/22, 5%) patient with metastatic osteosarcoma treated with pembrolizumab and no patients (n = 0/8) treated with escalating doses of ipilimumab [78]. There are currently ongoing small trials of pembrolizumab [79] and avelumab [80] in metastatic or resectable osteosarcoma whose results are expected in 2023 and are likely to guide future studies. It remains to be seen whether single agent immunotherapy alone will improve outcomes for patients, but likely a combined approach will be needed to overcome immunotherapy resistance. In vitro work by Ocadlikova et al. has demonstrated that cell lines treated with tyrosine kinase inhibitor sunitinib had increased PD-L1 expression and promoted immune cell activation [81], supporting a combined immunotherapy approach. Furthermore, the poor outcomes of single agent immunotherapy in clinical trials may be explained by an immune suppressive tumour microenvironment. Recent in vitro work to selectively inhibit immunosuppressive cells found in the microenvironment (myeloid-derived suppressor cells and tumour-associated macrophages) lead to a reprograming of the microenvironment to be more immune favourable [82]. This suggests a way forward may be to combine immunotherapy with drugs which promote an immune-activating microenvironment; specifically using this precision-based approach for patients with features of low or average immune infiltration. Finally, it is indeed possible that a chemoimmunotherapy may be more effective than either alone, as suggested in a recent case report of paclitaxel and nivolumab against Ewing sarcoma [83] and a large study of patients with metastatic triple negative breast cancer treated with nab-paclitaxel an atezolizumab [84].

## 8. The Role of Specialist Referral Centres

Finally, it is important to emphasize the benefit of having care at a specialised centre to facilitate multidisciplinary care. Particularly, surgery at high volume-specialised centres has been shown to increase OS in patients with other sarcomas [85]. European guidelines [11,86] recommend referral to a specialist bone or sarcoma centre for all patients. In the USA mifamurtide is available in a centre with an investigational pharmacy, and a physician willing do the regulatory work needs to obtain compassionate access. Given the rarity of osteosarcoma, management of patients at designated referral centres is key to maximising access to and participation in clinical trials and to develop therapeutic alliances to reduce toxicity and quality of life. Furthermore, given the peak incidence is in the 2nd–3rd decades of life, such centres often have adolescent and young adult support programs, which are important to provide holistic care for young patients with osteosarcoma.

## 9. Conclusions

Beyond MAP chemotherapy, few current therapies have a clinically meaningful impact for patients with osteosarcoma. An international, cooperative, and collaborative translational approach is required to augment current understanding and identify potential means to improve patient outcomes. Current evidence suggests that the biology of osteosarcoma is frustratingly complex [87] with heterogenous putative oncogenic mechanisms found in patients with osteosarcoma. Research using DNA and RNA analyses must continue to elucidate both patient- and subtype-specific molecular hallmarks. Such work is required to determine both predictive and prognostic immune and molecular biomarkers at diagnosis and during therapy for patients with osteosarcoma. Ultimately, this work may lead to precision medicine-based approaches for patients with osteosarcoma and hopefully lead to a clinically meaningful improvement in outcomes for patients with osteosarcoma.

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
