# Peer review of "Future Directions in the Treatment of Osteosarcoma"

_cells, 2021, doi:10.3390/cells10010172_

Round 1

Reviewer 1 Report

The paper is well written and very interesting; only a couple of minor editing are required.

1) The author should upgrade the introduction according to the last WHO classification (2020). For example, the definition of osteosarcoma (line 32) does not include the production of cartilaginous or fibrous tissue; osteoblastic osteosarcoma (line 42) is not more the "conventional" type; the higher stature of patients with osteosarcoma (line 36-37) is not reported as clinically relevant.

2) The author shoud correct in line 82: good response is 90% or greater (not only > 90%); the same in line 144.

Reviewer 2 Report

Based on the title, this review should be more focusing on molecular analysis and immune therapies. However, these two points are not deeply discussed throughout the whole section. Why osteosarcoma relatively resistant to immunotherapy?  It would be more helpful if the authors could include more animal model studies and discuss osteosarcoma's heterogeneity, especially the tumor microenvironment, to highlight osteosarcoma immunotherapy's future directions.

The introduction is wordy, and some details should move to the other sections. The authors should point out the outstanding questions and outline what this review will be focusing on to discuss.

The conclusions section is not exact; the authors should summarize what is known, the leading-edge research focuses on now, and the future directions.

Round 2

Reviewer 2 Report

The authors changed the title and now try to focus on the future directions of osteosarcoma treatment. Given that osteosarcoma's complexity and relatively few therapy choices, I have only one more suggestion regarding precision medicine. The authors could discuss personalized therapy or precision medicine in the genome and gene expression analysis section and immunotherapy section and add a few more sentences in the conclusion section.

Author Response

Dear Dr Emanuela Palmerini and Professor Lee Jeys,

On behalf of my co-authors, we thank the reviewers for their feedback and comments. We have made the requested changes and our responses are highlighted individually below.

Sincerely,

Dr Alannah Smrke 

Reviewer 2:

The authors changed the title and now try to focus on the future directions of osteosarcoma treatment. Given that osteosarcoma's complexity and relatively few therapy choices, I have only one more suggestion regarding precision medicine.

The authors could discuss personalized therapy or precision medicine in the genome and gene expression analysis section and immunotherapy section

  • We have added in an additional reference (Subbiah 2015) illustrating comprehensive molecular profiling of 2 patients with chemotherapy resistant osteosarcoma to further discuss this in the genome and gene expression analysis section.
  • We have added a specific discussion of precision medicine to introduce the second to last paragraph in this section to more clearly highlight the proposed future role of molecular profiling and precision medicine.
  • We have added a sentence in the immunotherapy section to highlight that immunotherapy combinations can be a precision-based approach for patients with low immune infiltration tumours.

and add a few more sentences in the conclusion section.

  • We have added two additional sentences to the conclusion, which focus on precision medicine and genome/gene expression analysis